# Thirteen Ovary-Enriched Genes Are Individually Not Essential for Female Fertility in Mice

**DOI:** 10.3390/cells13100802

**Published:** 2024-05-08

**Authors:** Anh Hoang Pham, Chihiro Emori, Yu Ishikawa-Yamauchi, Keizo Tokuhiro, Maki Kamoshita, Yoshitaka Fujihara, Masahito Ikawa

**Affiliations:** 1Research Institute for Microbial Diseases, Osaka University, Osaka 565-0871, Japan; anh-pham@biken.osaka-u.ac.jp (A.H.P.); emoric@biken.osaka-u.ac.jp (C.E.); kamoshita@biken.osaka-u.ac.jp (M.K.); fujihara@ncvc.go.jp (Y.F.); 2Graduate School of Pharmaceutical Sciences, Osaka University, Osaka 565-0871, Japan; 3Department of Regenerative Medicine, Yokohama City University Graduate School of Medicine, Yokohama 236-0027, Japan; ishikawa.yu.ym@yokohama-cu.ac.jp; 4The Institute of Medical Science, The University of Tokyo, Tokyo 108-8639, Japan; 5Department of Genome Editing, Institute of Biomedical Science, Kansai Medical University, Osaka 573-1191, Japan; tokuhirk@hirakata.kmu.ac.jp; 6Department of Advanced Medical Technologies, National Cerebral and Cardiovascular Center, Osaka 564-8565, Japan; 7Center for Infectious Disease Education and Research, Osaka University, Osaka 565-0871, Japan

**Keywords:** CRISPR/Cas9, female fertility, knockout mouse, fertilization

## Abstract

Infertility is considered a global health issue as it currently affects one in every six couples, with female factors reckoned to contribute to partly or solely 50% of all infertility cases. Over a thousand genes are predicted to be highly expressed in the female reproductive system and around 150 genes in the ovary. However, some of their functions in fertility remain to be elucidated. In this study, 13 ovary and/or oocyte-enriched genes (*Ccdc58*, *D930020B18Rik*, *Elobl*, *Fbxw15*, *Oas1h*, *Nlrp2*, *Pramel34*, *Pramel47*, *Pkd1l2*, *Sting1*, *Tspan4*, *Tubal3*, *Zar1l*) were individually knocked out by the CRISPR/Cas9 system. Mating tests showed that these 13 mutant mouse lines were capable of producing offspring. In addition, we observed the histology section of ovaries and performed in vitro fertilization in five mutant mouse lines. We found no significant anomalies in terms of ovarian development and fertilization ability. In this study, 13 different mutant mouse lines generated by CRISPR/Cas9 genome editing technology revealed that these 13 genes are individually not essential for female fertility in mice.

## 1. Introduction

Infertility is defined as an inability to achieve pregnancy after 12 months or more of regular unprotected sexual intercourse with a healthy partner [1]. Infertility is a global health issue as it currently affects one in every six couples worldwide [2], with female factors reckoned to contribute to partly or solely 50% of all infertility cases [3]. Approximately 85% of female infertile causes are categorized into five aspects: ovulatory disorders, endometriosis, pelvic adhesions, tubal blockage, uterine abnormalities, and hyperprolactinemia [4]. The remaining 15% are considered “unexplained infertility” in which the currently available diagnostic methods are unable to detect any abnormality in either individual [5].

It has been estimated that 50% of infertility cases are due to genetic defects. However, only a limited number of genes have been identified as directly causing or significantly linked to primary infertility [6], which is a significant gap in understanding how genes contribute to reproductive issues in women. Thus, it is crucial to uncover the genetic foundation of these conditions. Human transcriptome analysis revealed that thousands of genes are predicted to express in the ovary, and 178 genes are upregulated at least four-fold compared to those in other tissues [7], with about 20% of studied genes reported to be essential for female fertility. This suggests that investigating the function of genes that are highly expressed in ovaries or oocytes would be beneficial to understand the causes of female infertility. Due to the ethical concerns of using human samples for experimental studies, mice have been a well-used model for genetic studies because of their striking similarity with the human genome. Studying mouse models has revealed the mechanism of many female reproductive-enriched genes.

For example, the disruption of *Pabpn1l* (poly(A)-binding protein nuclear 1-like) has resulted in halted embryo development at the two-cell stage due to maternal mRNA degradation failure [8,9]. *Foxl2* (forkhead box L2), one of the well-known ovarian markers, has been characterized in many clinical and basic studies to be directly associated with blepharophimosis/ptosis/epicanthus inverse syndrome (BPES)—a symptom that leads to infertility in which the patients’ ovary function is prematurely terminated [10]. However, a considerable number of ovary-enriched genes have been unexpectedly found to be individually dispensable for female fertility, such as oocyte-secreted protein 1, 2, 3 (*Oosp1/2/3*) [11] and oocyte-specific linker histone *H1foo* [12]. As a result, gene knockout experiments have been considered a reliable approach to determine the exact function of targeted genes in vivo.

Thanks to the emergence of CRISPR/Cas9 genome editing technology that enables a rapid and cost-effective generation of knockout (KO) mice, large-scale genetic screening using these mice has been employed in multiple reproductive studies to identify hundreds of dispensable fertility genes [13,14]. Disseminating such information about fertile genes assists researchers by avoiding redundant efforts in generating identical KO mice or carrying out extensive in vitro experiments on genes that may be non-essential in vivo.

Here, we generated 13 different KO mouse lines by deleting targeted genes (*Ccdc58*, *D930020B18Rik*, *Elobl*, *Fbxw15*, *Oas1h*, *Nlrp2*, *Pramel34*, *Pramel47*, *Pkd1l2*, *Sting1*, *Tspan4*, *Tubal3*, *Zar1l)* that were bioinformatically predicted to be expressed predominantly in the ovary using the CRISPR/Cas9 system. Then, the essentiality of these target genes for female fertility was determined by the fecundity of the KO mouse.

## 2. Materials and Methods

### 2.1. Animals

The act of using animals and ethics in this study was approved by the Institutional Animal Care and Use Committees of the Research Institute for Microbial Diseases of Osaka University (Biken-AP-R03-01-1), the University of Tokyo (A21-28), Kansai Medical University (24-048, 23-010), and National Cerebral and Cardiovascular Center (24039, 23037). All the experiments involving animals were conducted following the guidelines and regulations for animal experimentation. B6D2F1 and ICR mice used in the laboratory were purchased from CLEA Japan, Inc. (Tokyo, Japan) and Japan SLC, Inc. (Shizuoka, Japan).

### 2.2. In Silico Gene Expression Analysis of Candidate Genes

To select candidate genes, we utilized bioinformatic databases of NCBI UniGene project (currently unavailable). We also used RNA sequence data of GV and MII oocytes performed in our *Papbn1l* KO mouse study [8] to find genes that are highly expressed in oocytes. Based on the Mammalian Reproductive Genetics database v2.0 (MRGDv2) (https://orit.research.bcm.edu/MRGDv2, accessed on 8 April 2024), we identified the expression profiles of the 13 genes (*Ccdc58*, *D930020B18Rik*, *Elobl*, *Fbxw15*, *Nlrp2*, *Oas1h*, *Pkd1l2*, *Pramel34*, *Pramel47*, *Sting1*, *Tspan4*, *Tubal3*, *Zar1l*). Transcripts per million (TPM) values were obtained from the MRGDv2, and Graphpad Prism version 9.0 was used for heatmap visualization. Within the program, the TPM values of different tissues of a gene were normalized so that the highest value equals one and the lowest equals zero.

### 2.3. Generation of Knockout Mice with the CRISPR/Cas9 System

All KO mouse lines in this study were generated using the CRISPR/Cas9 genome editing system. A set of guide RNAs (gRNAs) was designed using web tools CRISPRdirect (crispr.dbcls.jp) and CRISPOR (crispor.tefor.net). Based on the tools, we selected a set of gRNAs with high criteria scores and minimized off-target frequency (Appendix A). The crRNA and tracrRNA were purchased from Integrated DNA technologies (IDT, Coralville IA, USA). The crRNA and tracrRNA were diluted with nuclease-free water and subsequently denatured at 95 °C for 1 min and allowed to anneal into functional gRNAs by cooling at room temperature. The gRNAs were then mixed with TrueCut™ Cas9 Protein v2 (Thermo Fisher scientific, Waltham, MA, USA) solution at 37 °C for 5 min. The final concentrations of gRNA and Cas9 used were 100 ng/µL and 40 ng/µL, respectively. The ribonucleoprotein complex of crRNA/tracrRNA/Cas9 was then introduced into the B6D2F1 zygotes by electroporation using an NEPA21 Super Electroporator (NEPAGENE, Chiba, Japan). For zygote collection, WT B6D2F1 female mice were superovulated with a serial intraperitoneal injection of CARD HyperOva (2.5 units, Kyudo Co., Saga, Japan) [15] followed by human chorionic gonadotropin (hCG, 5 units, ASKA Animal Health Co, Tokyo, Japan) and caged with B6D2F1 males. Females with observable plugs the following morning were dissected and two-pronuclear (2-PN) zygotes were collected from their oviducts for electroporation. Electroporated embryos were cultured in potassium simplex optimization medium with amino acid (KSOMaa) medium [16] overnight to develop into 2-cell-stage embryos. Twenty embryos were transplanted into the oviducts of a 0.5-day pseudopregnant ICR mouse. After 19.5 days, the founder generation was obtained by natural delivery or cesarean section. The genomes of F0 generation mice were validated by genomic polymerase chain reaction (PCR). Heterozygous F1 generation mice whose deletion sites were confirmed by Sanger sequencing were mated with each other to obtain the F2 homozygous mouse. Genotyping was performed through PCR using primers listed in Appendix A. Some KO mice generated in this study will be available to other investigators through the RIKEN BioResource Research Center, Japan (https://web.brc.riken.jp/en/, accessed on 1 April 2024) (RBRC numbers; *Ccdc58* 11982, *D930020B18Rik* 11967, *Elobl* 11996, *Oas1h* 11970, *Pramel34* 11671, *Pramel47* 11672, *Pkd1l2* 11968, *Tubal3* 11171, *Zar1l* 11456), or Center for Animal Resources and Development (CARD), Kumamoto University, Kumamoto, Japan (http://cardb.cc.kumamoto-u.ac.jp/transgenic/, accessed on 1 April 2024) (CARD ID: *Ccdc58*: 3351, *D930020B18Rik*: 3336, *Elobl*:3365, *Oas1h*: 3339, *Pkd1l2*: 3337, *Zar1l*: 3100).

### 2.4. Fertility Analysis of KO Lines

Sexually mature wild-type (WT) or KO female mice (8–20 weeks old) were caged individually with an eight-week-old WT male for at least eight weeks. Male mice were removed after the mating period, and females were kept for another three weeks to count the final litters. The number of pups and litters were counted every morning.

### 2.5. Histological Analysis of Ovaries

We collected ovaries from mice to be used for in vitro fertilization (IVF) experiments in order to minimize the number of mice that had to be sacrificed for the experiment. We could still evaluate whether ovulation occurred and whether follicles developed properly. A forty-eight-hour interval serial injection of pregnant mare serum gonadotropin (PMSG, 5 units, ASKA Animal Health Co, Tokyo, Japan) followed by hCG (5 units) was administered into the abdominal cavity of eight to twelve-week-old female mice. A total of 14 h after the hCG administration, ovaries were dissected from the female mice, fixed in Bouin’s solution (Polysciences, Warrington, PA, USA), embedded in paraffin wax, and sectioned at a thickness of 5 µm on a Microm HM325 microtome (Microm, Walldorf, Germany). PAS staining was performed to evaluate the ovary morphology. In brief, ovary sections were stained with 1% periodic acid (Nacalai Tesque, Kyoto, Japan) followed by treatment of Schiff’s reagent (FUJIFILM Wako, Osaka, Japan). The sections were finally counterstained with Mayer’s hematoxylin solution (FUJIFILM Wako, Osaka, Japan) before being observed under an Olympus BX53 microscope equipped with an Olympus DP74 color camera (Olympus, Tokyo, Japan).

### 2.6. In Vitro Fertilization (IVF)

Cauda epididymal spermatozoa extracted from sexually mature wild-type males were preincubated in the 100 µL TYH medium drops [17] for 2 h at 37 °C, 5% CO_2_ to induce capacitation. Cumulus–oocyte complexes (COCs) were extracted from the oviductal ampulla of eight to twelve-week-old female mice after ovulation was induced by injecting PMSG followed by hCG. COCs were then incubated with spermatozoa at 2 × 10^5^ sperm/mL in 100 µL TYH medium drops for 4 h. COCs were then treated with 300 µg/mL of hyaluronidase (FUJIFILM Wako, Osaka, Japan) for 5 min to remove the cumulus cells. Fertilized embryos were cultured in KSOMaa medium drops until the Blastocyst stage to record the developmental rate. We counted 2-cells to elucidate the fertilization rate the day after insemination. Blastocysts were counted at three days after insemination.

### 2.7. Statistical Analyses

Statistical difference was determined using Welch’s *t*-test using Microsoft Office Excel (Microsoft Corporation, Redmond, WA, USA). Differences were considered statistically significant if the *p* values were less than 0.05. Data represent the mean ± standard deviation (SD).

## 3. Results

### 3.1. Expression Patterns of Candidate Genes in Mice

We selected candidate genes based on the UniGene database and our RNA seq data, as described in the Materials and Methods section. A heatmap illustrating the ovary/oocyte-enriched expression pattern of these 13 genes was also generated based on the data obtained from the MRG database (Appendix A). *D930020B18Rik, Elobl*, *Fbxw15*, *Nlrp2*, *Oas1h*, and *Zar1l* exhibited robust expression, particularly prominent in the early antral follicles and GV-stage oocytes. The high expression levels of *Pramel34* and *Tubal3* during the MII-stage oocytes indicate their potential to be important factors for the fertilization capability of oocytes.

### 3.2. Fertility Results of 13 Ovary-Enriched KO Mouse Models

KO female mice were caged with a WT male for at least eight weeks to examine the fecundity of the KO female mice at their sexually mature ages. In total, 12 KO strains (*Ccdc58*, *D930020B18Rik*, *Elobl*, *Fbxw15*, *Oas1h*, *Pramel34*, *Pramel47*, *Pkd1l2*, *Sting1*, *Tspan4*, *Tubal3*, *and Zar1l)* showed an equivalent number of pups/litter compared to WT. We observed a significant decrease in pups/litter of *Nlrp2* KO females (5.4 ± 2.6, *p* < 0.05), which had already been reported in previous studies [18,19] (Table 1).

In addition to the fertility test, we examined the ovarian morphology and in vitro fertilization ratio (Table 2 and Table 3) of five genes (*Ccdc58*, *D930020B18Rik*, *Elobl*, *Oas1h*, *Pkd1l2*) to check whether there were any differences between KO and WT.

### 3.3. Phenotypic Analysis of Ccdc58 KO Mouse Line

*Ccdc58* (also called Mix23, mitochondrial matrix import factor 23) encodes a protein carrying a coiled-coil domain (Figure 1A). *Ccdc58*^−/−^ mice were generated using two gRNAs designed to cover the upstream of exon 2 and downstream of the stop codon in exon 5 (Figure 1B). The mutant alleles were confirmed by genomic PCR (Figure 1C) using specific primers shown in Appendix A. Sanger sequencing showed that the *Ccdc58*^−/−^ mouse had a 9893 bp deletion (Figure 1C). We first observed the histology of the ovary after hCG injection following PMSG. PAS staining of the *Ccdc58*^+/−^ and *Ccdc58*^−/−^ ovary sections revealed no observable differences in ovary morphology (Figure 1D). We then performed IVF using *Ccdc58*^−/−^ or *Ccdc58*^+/−^ MII oocytes and WT sperm. The ratio of 2-cell-stage embryos derived from *Ccdc58*^−/−^ mice was 81.2 ± 20.9%, which was comparable with that of *Ccdc58*^+/−^ (89.9 ± 10.0%) (Table 2, Figure 1E). Additionally, there were no significant differences in the blastocyst rates of *Ccdc58*^−/−^ embryos and *Ccdc58*^+/−^ (64.7 ± 16.0% and 79.0 ± 15.9%, respectively) (Table 3, Figure 1F).

### 3.4. Phenotypic Analysis of D930020B18Rik KO Mouse Line

*D930020B18Rik* (RIKEN cDNA D930020B18 gene) encodes a protein with an unknown functional domain named DUF4551 (Figure 2A). The whole protein coding sequence of *D930020B18Rik* was disrupted by the CRISPR/Cas9 system (Figure 2B). Genomic PCR followed by Sanger sequencing verified that the mutant mouse line carried a 50,969 bp deletion (Figure 2C). Ovary histology results showed no perceptible differences between the KO mouse and the control heterozygous mouse (Figure 2D). IVF results showed that the ratio of *D930020B18Rik*^−/−^ 2-cell-stage embryos (86.1 ± 12.5%) was comparable to that of *D930020B18Rik*^+/−^ (88.4 ± 8.6%) (Table 2, Figure 2E). No significant differences were observed in the blastocyst ratio between *D930020B18Rik*^−/−^ (83.1 ± 9.9%) and *D930020B18Rik*^+/−^ (81.9 ± 6.3%) (Table 3, Figure 2F).

### 3.5. Phenotypic Analysis of Elobl KO Mouse Line

ELOBL (elongin B-like) comprises a ubiquitin-like (Ubl) domain (Figure 3A), suggesting a possibility that this protein has a specific function in the proteasome. We designed a set of gRNAs that targeted the whole gene protein coding region (Figure 3B). Genomic PCR was performed to identify the mutant mouse using primers presented in Appendix A. The resulting mutant allele carrying a 2066 bp deletion was confirmed by Sanger sequencing analysis (Figure 3C). After the hormonal injection of PMSG and hCG, ovaries were collected for histology. *Elobl*^−/−^ ovaries exhibited no obvious abnormalities compared to the *Elobl*^+/−^ (Figure 3D). IVF experiments indicated no significant difference in the 2-cell stage embryo rate between *Elobl*^−/−^ (88.8 ± 8.2%) and *Elobl*^+/−^ (86.3 ± 13.7%) (Table 2, Figure 3E) as well as the blastocyst rate (78.8 ± 12.2% and 80.1 ± 19.4%, respectively) (Table 3, Figure 3F).

### 3.6. Phenotypic Analysis of Oas1h KO Mouse Line

*Oas1h* (2′-5′ oligoadenylate synthetase 1H) encodes a protein with a polymerase nucleotidyltransferase domain and a 2′-5′-oligoadenylate synthetase 1 domain (Figure 4A). The whole protein-coding region of *Oas1h* was deleted using a set of gRNAs flanking both the gene start codon and end codon (Figure 4B). The mutant site was verified as a deletion of 11,894 bp (Figure 4C). We did not find any significant difference between the *Oas1h*^−/−^ and *Oas1h*^+/−^ ovary histology (Figure 4D). IVF experiments showed no significant difference in the ratio of *Oas1h*^−/−^ oocytes and *Oas1h*^+/−^ oocytes that developed to the 2-cell stage (92.2 ± 10.6% and 92.6 ±12.8%, respectively) (Table 2, Figure 4E). However, the ratio of blastocyst-stage embryos was significantly lower in the *Oas1h*^−/−^ (66.6 ± 15.0%) compared to *Oas1h*^+/−^ (92.6 ± 12.8%) (Table 3, Figure 4F).

### 3.7. Phenotypic Analysis of Pkd1l2 KO Mouse Line

As a member of the TRPP family protein, PKD1l2 (polycystic kidney disease 1 like 2) is a cation protein comprising six transmembrane domains, eight topological domains, and one intramembrane interleaving with each other (Figure 5A). The majority of the *Pkd1l2* protein-coding sequence was deleted, and genomic PCR followed by Sanger sequencing revealed a 72,716 bp deletion (Figure 5B). Histological analysis revealed no obvious abnormality in the ovary of *Pkd1l2*^−/−^ mice and *Pkd1l2*^+/−^ mice (Figure 5C). IVF showed no significant differences in the 2-cell stage embryo ratio between *Pkd1l2*^−/−^ (92.2 ± 5.9%) and *Pkd1l2*^+/−^ (93.3 ± 4.2%) (Table 2, Figure 5E) as well as the blastocyst ratio (*Pkd1l2*^−/−^ 85.8 ± 7.6% and *Pkd1l2*^+/−^ 90.0 ± 7.0%) (Table 3, Figure 5F).

## 4. Discussion

In this study, we examined the essentiality of 13 genes in female reproduction by generating knockout mouse lines using the CRISPR/Cas9 system. The knockout females exhibited normal fecundity, suggesting that these 13 genes are individually not essential for female fertility even though the 13 genes in our study are highly expressed in the ovary or oocytes.

One hypothesized reason why 13 KO strains are fertile is that there might be other family genes that compensate for their specific function. For instance, *Oas1h* has eight other family genes (*Oas1a-h*) due to an extensive gene amplification of OAS1 in rodents. Like the compromised blastocyst rate found from our IVF result of the *Oas1h* KO mouse, a previous study reported that *Oas1d* KO mice exhibited defects in preimplantation development [20]. Since both *Oas1d* and *Oas1h* KO mice showed a similar phenotype but still maintained their fertility, they may have a similar function. In addition, all the members of Oas1 family genes are located next to each other on chromosome 5 [21]. There is a possibility that these *Oas1* family members participate in early embryo development and their roles are similar enough to compensate for each other. A previous review article from our group showed that 1262 genes are highly expressed in the mouse testis in which 549 genes are phenotypically annotated. Among them, 242 genes were indicated to be infertile or have reduced fertility [22]. Even though there are many testis-specific genes, about 60% of them are not individually essential for male fertility based on single KO phenotypic analysis. This can also explain why we did not observe any minor phenotype in our KO mouse lines.

Several genes in this study have been well characterized in vitro and have been reported previously. *Ccdc58* is expressed in multiple tissues, including the ovary, and has been well characterized in yeast cells as one of the mitochondrial matrix import factors under TIM23-induced stress conditions [23]. In addition, high through-put interaction studies revealed that CCDC58 could interact with various proteins in the mitochondrial intermembrane space, such as ATPase family gene 3 like 2 (AFG3L2), apoptosis-inducing factor mitochondria-associated 1 (AIFM1), and cytochrome c [24,25]. Since mitochondria are one of the most important organelles that determine the quantity and quality of oocytes [26], exploring the role of *Ccdc58* would be beneficial, although our results showed that *Ccdc58* is not essential for female fertility.

*Elobl* has been identified as an ovary-enriched gene formed from a pseudogene insertion and subsequently recycled into a protein-coding gene by a retrotransposon insertion. Our study revealed that the disruption of *Elobl* did not affect the KO female mouse fecundity. There is a possibility that a family gene called *Elob* compensated for the loss of function caused by *Elobl* KO. The expression of *Elobl* was found to increase slightly in *Sirena1* KO oocytes, implying its potential involvement in an unusual mitochondrial distribution that was observed in the *Sirena1* KO oocytes [27]. However, the ATP production affected by such abnormal mitochondrial distribution could be compensated by surrounding cumulus cells because of gap junctions through which ATP can diffuse [28].

We observed that the blastocyst rate derived from *Oash1* KO oocytes decreased compared to the wild type. In vitro conditions may exert some adverse effects on the developing embryo compared to a natural in vivo system, which could explain why we observed the defect only during IVF but not in the litter size. Perhaps *Oas1h*, while not essential for KO mouse fecundity, still contributes to the preimplantation development process since we recorded a decrease in the blastocyst rate but not in the 2-cell rate. Nevertheless, further studies of the whole *Oas1* family genes are required to reveal their role in female fertility at a molecular level.

PKD1L2 is a member of the TRPP family, typically featured as a large group of cation proteins comprising multiple transmembrane domains [29]. PKD1L2 is predicted to function as an ion channel in vivo as a result of being a member of the TRPP family [30] and has been reported to be able to bind to specific G-protein subunits in vitro [31]. *PKD1L2* expression was detected to be among the most downregulated in the adipose tissue of polycystic ovary syndrome (PCOS) patients [32], suggesting that *Pkd1l2* could function as a potential biomarker for PCOS diagnosis. In our study, *Pkd1l2* KO females had normal fecundity, implying that the disruption of *Pkd1l2* individually would not detrimentally affect female fertility.

As Peng et al. [33] indicated, *Nlrp2* mRNA was restricted solely to the oocyte. We also generated our own *Nlrp2* KO mice and confirmed that *Nlrp2* KO females had decreased fertility, supporting the previous study [16,17]. In these studies, the role of *Nlrp2* in female fertility has been elucidated extensively, and NLRP2 has been demonstrated to be a part of the subcortical maternal complex (SCMC). Hence, the loss of maternal NLRP2 would severely affect the localization of SCMC protein members, including the core protein TLE6, leading to subfertility, follicular atresia, and abnormal cleavages during embryogenesis [18]. Additionally, *Nlrp2* was reported to function as a critical checkpoint, gaining significance with the increase in maternal age. *Nlrp2* KO female mice exhibited a decreased litter size only when they were older than 15 weeks, but not at 8–15 weeks old [19]. Since our study focused on analyzing mice that are at sexually mature ages (8–20 weeks old age), we did not observe any effects of aging on the reproductive life span in the KO mouse line. If we extended the length of mating, we may have observed a declining litter size with age in our KO mouse lines due to an impaired reproductive lifespan.

*Zar1l* is considered to hold the potential as an essential factor for female fertility because of its high homology in the C-terminal RNA-binding domain with *Zar1*, an essential gene that regulates the maternal–zygotic transition (MZT). The underlying molecular mechanism and the role of *Zar1l* in oocyte meiotic maturation have been well characterized to only strengthen the *Zar1* infertile phenotype in another study [34]. In our study, we confirmed that *Zar1l* KO females are fertile, which is consistent with the previous report.

## 5. Conclusion

In the present study, we successfully generated 13 different KO mouse lines and validated their non-essential roles in fertility via a mating test in addition to a phenotypical analysis of five different KO mouse lines. While it is undeniable that these 13 genes could play a role in significant reproductive events when combined with other factors or have minor phenotypes that do not result in female infertility, their individual redundancy reduces their clinical importance and makes them less viable as subjects for studies on female fertility. Overall, our in vivo analysis of ovary-enriched genes has demonstrated an approach for screening genetic factors for female fertility in which a single loss-of-function mutation would severely jeopardize normal female reproduction.

## Figures and Tables

**Figure 1 cells-13-00802-f001:**
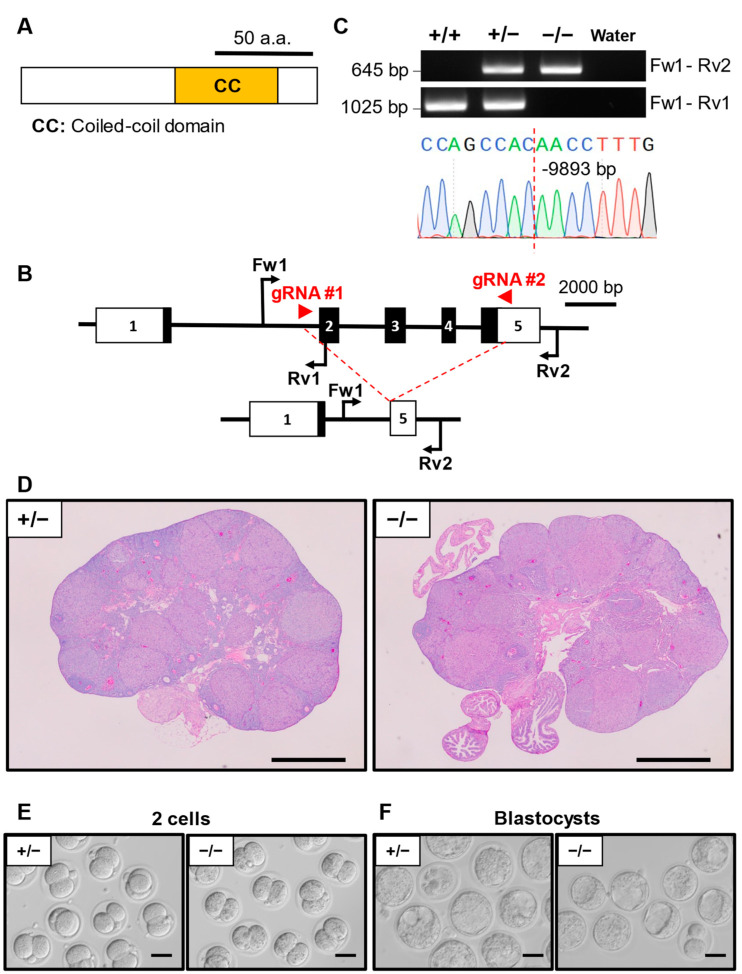
Phenotypic analysis of *Ccdc58* KO female mice. (**A**) Protein domain of CCDC58. CC: coiled-coil domain. (**B**) KO strategy of *Ccdc58*. Black boxes indicate exons; white boxes indicate untranslated regions. Two gRNAs were designed to target exon 2 and exon 5. Fw1: forward primer for the genotyping. Rv1/2: reverse primers for genotyping. (**C**) Genotyping PCR result of *Ccdc58* KO mice and subsequent Sanger sequencing. (**D**) Histological analysis of ovary in *Ccdc58*^+/−^ and ^−/−^. Scale bar = 500 µm. (**E**) Representative pictures of 2-cell embryos after IVF derived from *Ccdc58^+/^*^−^ or ^−/−^ MII oocytes and WT sperm. Scale bar = 50 µm. (**F**) Representative pictures of blastocyst embryos after IVF derived from *Ccdc58*^+/−^ or ^−/−^ MII oocytes and WT sperm. Scale bar = 50 µm.

**Figure 2 cells-13-00802-f002:**
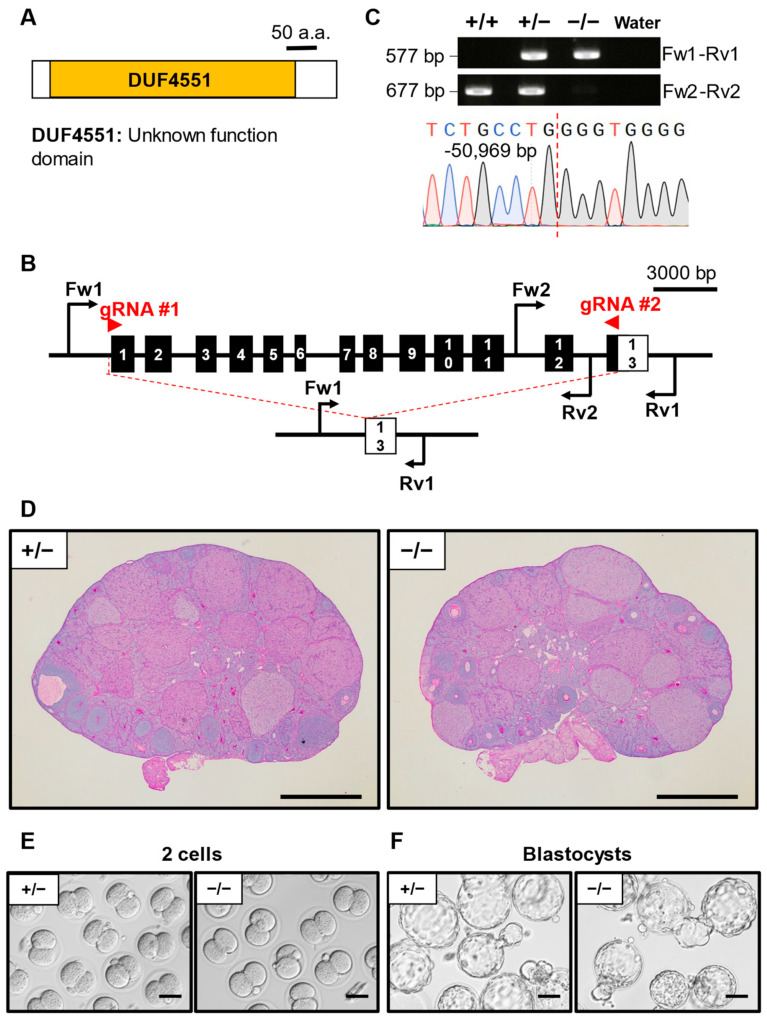
Phenotypic analysis of *D930020B18Rik* KO female mice. (**A**) Protein domain of D930020B18RIK. DUF4551: unknown functional domain. (**B**) KO strategy of *D930020B18Rik*. Black boxes indicate exons; the white box indicates an untranslated region. Two gRNAs were designed to target exon 1 and exon 13. Fw1/2: forward primer for genotyping. Rv1/2: reverse primers for genotyping. (**C**) Genotyping PCR result of *D930020B18Rik* KO mice and subsequent Sanger sequencing. (**D**) Histological analysis of ovary in *D930020B18Rik*^+/−^ and ^−/−^. Scale bar = 500 µm. (**E**) Representative pictures of 2-cell embryos after IVF derived from *D930020B18Rik*^+/−^ or ^−/−^ MII oocytes and WT sperm. Scale bar = 50 µm. (**F**) Representative pictures of blastocyst embryos after IVF derived from *D930020B18Rik*^+/−^ or ^−/−^ MII oocytes and WT sperm. Scale bar = 50 µm.

**Figure 3 cells-13-00802-f003:**
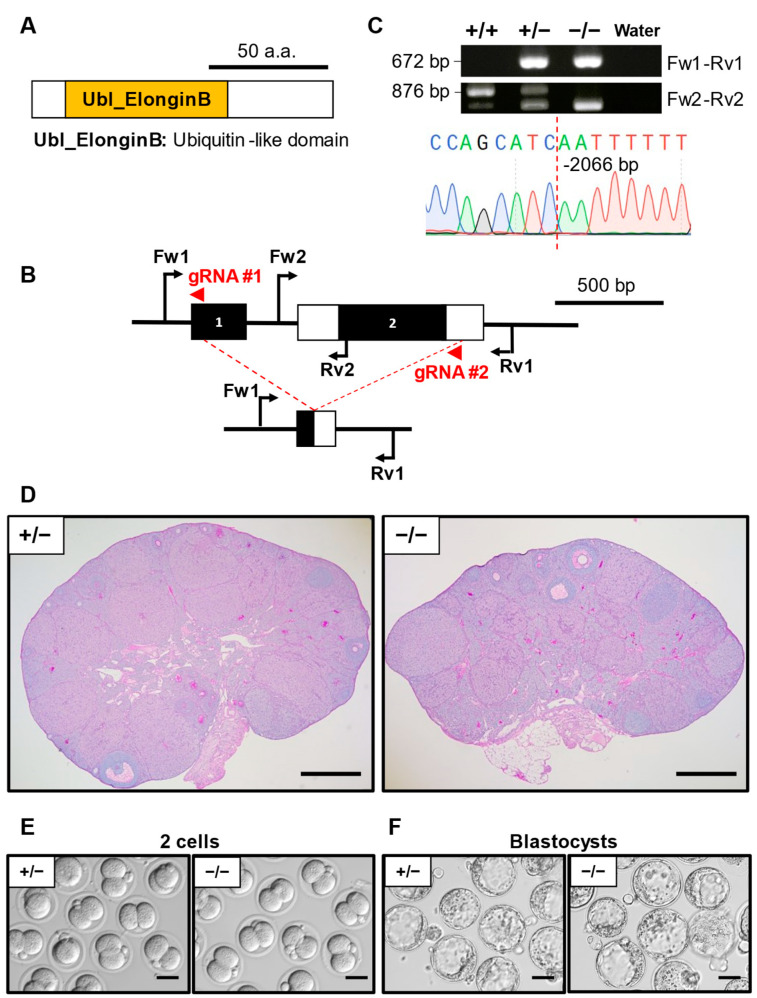
Phenotypic analysis of *Elobl* KO female mice. (**A**) Protein domain of ELOBL. Ubl ElonginB: ubiquitin-like domain. (**B**) KO strategy of *Elobl*. Black boxes indicate exons; white boxes indicate untranslated regions. Two gRNAs were designed to target exon 1 and exon 2. Fw1/2: forward primer for genotyping. Rv1/2: reverse primers for genotyping. (**C**) Genotyping PCR result of *Elobl* KO mice and subsequent Sanger sequencing. (**D**) Histological analysis of ovary in *Elobl*^+/−^ and ^−/−^. Scale bar = 500 µm. (**E**) Representative pictures of 2-cell embryos after IVF derived from *Elobl*^+/−^ or ^−/−^ MII oocytes and WT sperm. Scale bar = 50 µm. (**F**) Representative pictures of blastocyst embryos after IVF derived from *Elobl*^+/−^ or ^−/−^ MII oocytes and WT sperm. Scale bar = 50 µm.

**Figure 4 cells-13-00802-f004:**
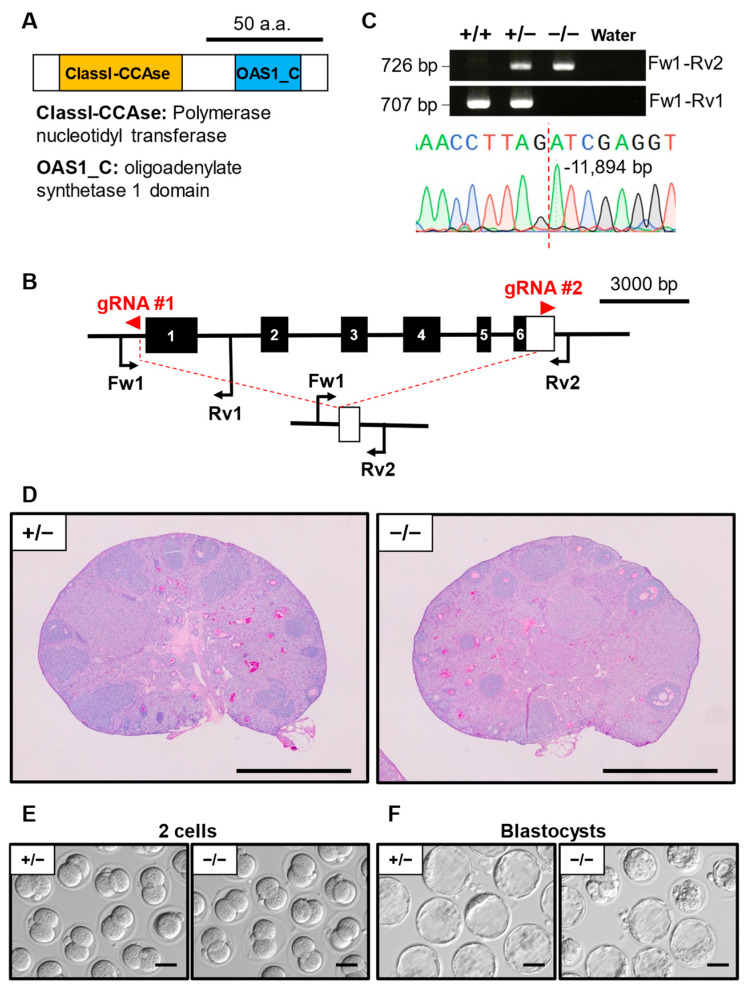
Phenotypic analysis of *Oas1h* KO female mice. (**A**) Protein domains of OAS1H. ClassI-CCAse: polymerase nucleotidyl transferase. OAS1_C: oligoadenylate synthetase 1 domain. (**B**) KO strategy of *Oas1h*. Black boxes indicate exons; the white box indicates an untranslated region. Two gRNAs were designed to target exon 1 and exon 6. Fw1: forward primer for the genotyping. Rv1/2: reverse primers for genotyping. (**C**) Genotyping PCR result of *Oas1h* KO mice and subsequent Sanger sequencing. (**D**) Histological analysis of ovary in *Oas1h*^+/−^ and ^−/−^. Scale bar = 500 µm. (**E**) Representative pictures of 2-cell embryos after IVF derived from *Oas1h*^+/−^ or ^−/−^ MII oocytes and WT sperm. Scale bar = 50 µm. (**F**) Representative pictures of blastocyst embryos after IVF derived from *Oas1h*^+/−^ or ^−/−^ MII oocytes and WT sperm. Scale bar = 50 µm.

**Figure 5 cells-13-00802-f005:**
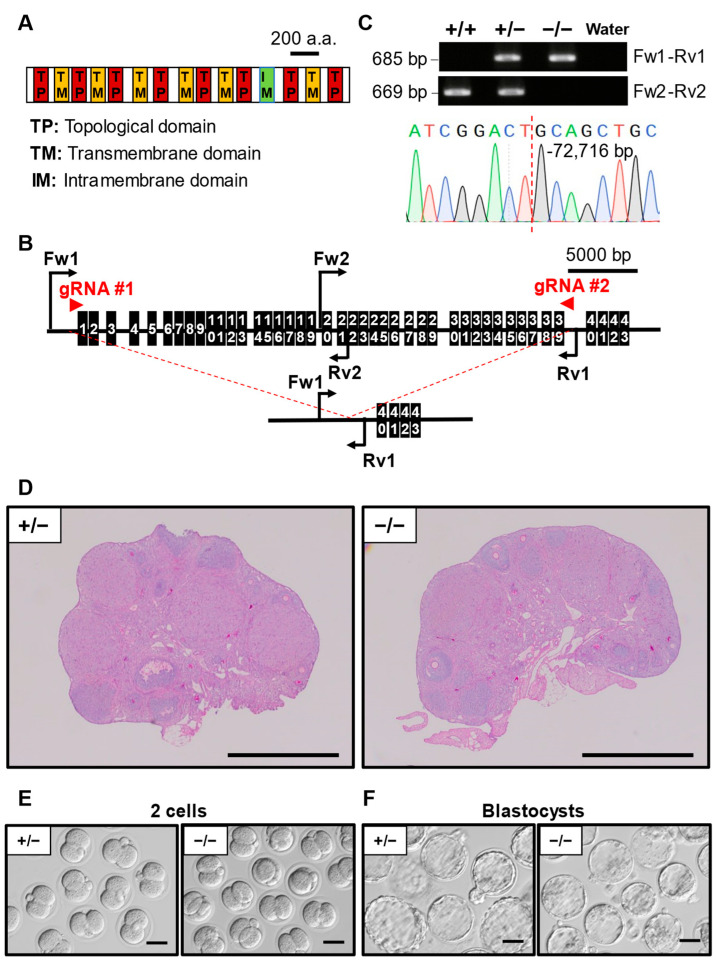
Phenotypic analysis of *Pkd1l2* KO female mice. (**A**) Protein domains of PKD1L2. TP: Topological domain. TM: Transmembrane domain. IM: Intramembrane domain. (**B**) KO strategy of *Pkd1l2*. Black boxes indicate exons. Two gRNAs were designed to target exon1 and exon39. Fw1/2: forward primer for genotyping. Rv1/2: reverse primers for genotyping. (**C**) Genotyping PCR result of *Pkd1l2* KO mice and subsequent Sanger sequencing. (**D**) Histological analysis of ovary in *Pkd1l2*^+/−^ and ^−/−^. Scale bar = 500 µm. (**E**) Representative pictures of 2-cell embryos after IVF derived from *Pkd1l2*^+/−^ or ^−/−^ MII oocytes and WT sperm. Scale bar = 50 µm. (**F**) Representative pictures of blastocyst embryos after IVF derived from *Pkd1l2*^+/−^ or ^−/−^ MII oocytes and WT sperm. Scale bar = 50 µm.

**Table 1 cells-13-00802-t001:** Female fertility of the 13 KO mouse lines. Statistical analysis of litter size between WT and each KO mouse strain was performed by Welch’s *t*-test. *; *p* < 0.01.

Gene Symbol	Genotype	No. of Females	No. of Pups	No. of Litters	Average Litter Size ± SD
	Wildtype	9	247	30	8.2 ± 2.7
*Ccdc58*	−9893/−9893	7	182	23	7.9 ± 3.1
*D930020B18Rik*	−50,969/−50,969	5	107	15	7.1 ± 2.4
*Elobl*	−2066/−2066	4	134	16	8.4 ± 3.0
*Fbxw15*	−15,610/−15,610	3	84	9	9.3 ± 1.1
*Nlrp2*	−9171/−9171	5	81	15	5.4 ± 2.6 *
*Oas1h*	−11,894/−11,894	3	85	10	8.5 ± 2.9
*Pkd1l2*	−72,716/−72,716	3	61	9	6.8 ± 1.9
*Pramel34*	−3037 + 20ins/−3037 + 20ins	3	131	16	8.2 ± 1.6
*Pramel47*	−3616/−3616	5	198	25	7.9 ± 2.2
*Sting1*	−5916/−5916	5	98	15	6.5 ± 2.9
*Tspan4*	−10,374/−10,374	4	119	17	7.0 ± 3.4
*Tubal3*	−3947/−3947	6	183	24	7.6 ± 1.8
*Zar1l*	−11,036/−11,036	3	77	11	7.0 ± 2.0

**Table 2 cells-13-00802-t002:** The 2-cell embryo rate after in vitro fertilization in 8 KO mouse lines. Statistical analysis of the 2-cell rate between each homozygous mouse strain and its heterozygous group was performed using Welch’s *t*-test. There were no significant differences found.

**Gene symbol**	2 cell-rate (%) ± SD, N = number of mice, n = number of total oocytes	
	Het	Hom
*Ccdc58*	89.9 ± 10.0, N = 5, n = 69	81.2 ± 20.9, N = 6, n = 62
*D930020B18Rik*	88.4 ± 8.6, N = 6, n = 136	86.1 ± 12.5, N = 5, n = 109
*Elobl*	86.3 ± 13.7, N = 6, n = 106	88.8 ± 8.2, N = 7, n = 171
*Oas1h*	92.6 ± 12.8, N = 3, n = 49	92.2 ± 10.6, N = 5, n = 125
*Pkd1l2*	93.3 ± 4.2, N = 6, n = 97	92.2 ± 5.9, N = 7, n = 147
*Pramel34*	82.5 ± 27.7, N = 4, n = 157	91.9 ± 4.9, N = 5, n = 187
*Pramel47*	89.2 ± 3.3, N = 4, n = 56	81.2 ± 14.5, N = 7, n = 116
*Tubal3*	80.3 ± 17.2, N = 5, n = 101	87.0 ± 10.8, N = 7, n = 162

**Table 3 cells-13-00802-t003:** Blastocyst-stage embryo rate after in vitro fertilization in 5 KO mouse lines. Statistical analysis of the blastocyst rate between each homozygous mouse strain and its heterozygous group was performed using Welch’s *t*-test. *; *p* < 0.05.

**Gene symbol**	Blastocyst rate (%) ± SD, N = number of mice, n = number of total oocytes	
	Het	Hom
*Ccdc58*	79.0 ± 15.9, N = 5, n = 69	64.7 ± 16.0, N = 6, n = 62
*D930020B18Rik*	81.9 ± 6.3, N = 6, n = 136	83.1 ± 9.9, N = 5, n = 109
*Elobl*	80.1 ± 19.4, N = 6, n = 106	78.8 ± 12.2, N = 7, n = 171
*Oas1h*	92.6 ± 12.8, N = 3, n = 49	66.6 ± 15.0 *, N = 5, n = 125
*Pkd1l2*	90.0 ± 7.0, N = 6, n = 97	85.8 ± 7.6, N = 7, n = 147

## Data Availability

Data are contained within the article.

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
