# Peer review of "Thirteen Ovary-Enriched Genes Are Individually Not Essential for Female Fertility in Mice"

_cells, 2024, doi:10.3390/cells13100802_

Round 1
Reviewer 1 Report
Comments and Suggestions for Authors
The authors point out that a large proportion of infertility among humans are due to genetic defects, and that our understanding of the mechanisms behind are limited as many of the genes involved are still not known. In this paper the authors provide evidence for the selection of 13 ovary expressed genes not to contribute to reproductive issues in women.
The authors use relevant and highly advanced methods to test selected genes for their potential to reduce fertility in mice when knocked out. The methods are solid and the results obtained are well documented and seem conclusive. The presentation of methods, results and discussion are well formulated and clear. The results are negative when it comes to detect relevant genes contributing to infertility, however should be published as the results are highly relevant for researchers in this field.
A main issue with this paper is the procedure used to select target genes. The number of candidate genes should be very large as all processes from sex differentiation, through ovary cell differentiation, meiosis, ovary formation and maturation may have impact on fertility. The authors explain that the 13 genes selected “were bioinformatically predicted», and have been checked in the MRG database. The actual procedure to select genes are not well descried. What was the criteria for selecting the target genes. This should be explained, with links to bioinformatics platforms used.
It would have been of interest if a gene/s with known negative effect on fertility were included in the study, but too late to include now. In Results, a heat map of the expression of the selected genes has been presented, which is ok. Is it possible to include data for some well-known, relevant genes like foxl2 and zrp 2 and 3. This may help as it could calibrate the expression pattern as selection method for genes to be included in KO experiments.
Author Response
 We would like to thank reviewer’s comments. Below are the answers to the comments.
 Regarding how we select candidate genes, we used NCBI Unigene to select genes that are highly expressed in ovary and/or oocytes. Unfortunately, this is not currently available (closed on 2019), Therefore we used Mammalian Reproductive Databases Ver2.0 (MRGDv2, https://orit.research.bcm.edu/MRGDv2) to check the expression pattern. We also checked our own RNA seq data for oocyte expressed genes. We added some explanation how we selected candidate genes in materials and methods (Line 96-105)
 Thank you for your suggestion regarding Figure S1. We added Foxl2 and Zp2 to compare gene expression patterns in Figure S1. We confirmed Foxl2, a granulosa cell marker, is expressed in ovary or cumulus cells but not in oocytes or zygotes, and Zp2 is expressed in oocytes and not in other somatic cells.
Reviewer 2 Report
Comments and Suggestions for Authors
In this study, Pham et al. generated 13 different knockout mutant mouse lines by deleting various ovary and/or oocyte-enriched genes using CRISPR/Cas9 genome editing technology. The mating test results showed that 12 KO stains have comparable litter size compared with wild-type mice. Additionally, they found that Nlrp2 KO mice had decreased litter size. Moreover, they performed histology and in vitro fertilization for 5 mutant mouse lines to examine the ovarian morphology and fertilization ability. No obvious ovarian morphology difference was found in all 5 mutant mouse lines.
Major concern:
1. The descriptions in the title and abstract are not accurate. Based on the mating test, the Nlrp2 KO mice had compromised litter size. Moreover, based on the in vitro fertilization assay, the Oas1h KO mice had compromised blastocyst rate. Therefore, not all tested 13 genes are dispensable for female fertility in mice. Most importantly, more experiments should be performed to support the current conclusions. For instance, the authors should perform follicle counting to examine the effects of gene KO on ovarian reserve and follicle development. Additionally, the authors should also tract the reproductive lifespan of KO mice to support their conclusions.
2. For the ovarian histology experiment, why did the authors perform PMSG and hCG injection first? Please clarify the reason behind this.
3. Lines 233-235: the ratio of blastocyst-stage embryos was significantly decreased in the Oas1h KO mice compared to Oas1h+/- mice. However, the Oas1h KO did not affect the litter size. The authors should discuss the potential reasons in the discussion.
Minor concern:
1. Lines 149-150, the authors state that they examined the in vitro fertilization ratio for 5 genes. However, they reported 2-cell embryo rate after IVF in 8 KO mouse lines in Table 2, which seems inconsistent. Please clarify this discrepancy.
2. Line 166, “Table 1F” should be corrected to “Figure 1F”.
3. In Tables 2 and 3, the total number of oocytes in each group should be reported.
Author Response
 We really appreciated reviewer’s comments. Below are the answers to the comments.
 Major concern:
- Thank you for pointing out such critical points. We changed “dispensable” to “not essential”. Considering reviewer’s comment, we thought using “not essential genes” would be more appropriate since all the KO mice could still be pregnant and give birth to healthy pups, although Nlrp2 KO female mice exhibited fewer pups per litter. We also changed some sentences in abstract. Nlrp2 and Oas1h KO mice showed some phenotype but still they can produce pups at least.
 In this study, we focused on KO mice that were at their sexually mature ages (8-20 weeks old). Therefore, we decided to examine ovarian histology and in vitro fertilization in addition to mating test. From that point, we concluded 13 genes are not essential for female fertility. We agree with the reviewer’s point. If we examine ovarian reserve or reproductive lifespan, we might see the effect of KO in some mouse lines. In fact, Nlrp2 KO mice showed normal fertility at 9-15 weeks old age and only exhibited decreased fertility as the mice got old (16-25 weeks and 26-45 weeks). We added some discussion about it (Line 470-476).
 2. To minimize the number of mice that had to be sacrificed for the experiment, we collected ovaries from mice to be used for IVF experiments. We believed that this is acceptable and reasonable because we can confirm the KO mice ovulate eggs appropriately and we can still check whether there are follicles developed. We added sentences to explain that (Lines 161-163).
 3. We are grateful for your suggestion to enrich our discussion part. We have been searching for further information and try to propose a hypothesis for such a phenomenon. One hypothesis is that the in vitro system could not provide an optimal condition like the in vivo system to support the developing embryos. To clarify this point, recovering embryos from KO or WT females at 3.5- or 4.5-days gestation and comparing their development rates may give us some idea. However unfortunately we currently did not have enough number of mice to analyze, we would like to leave that point for future experiments. We added several sentences to it in the discussion part (lines 434-453).
 Minor concern:
 1. We are grateful for your suggestion. Since the COVID outbreak was so severe during the time we performed the IVF experiments. We refrain from coming to the lab during that time and we were not able to record the developmental ratio for the 3 mouse lines: Pramel34, Pramel47 and Tubal3. We also examined the average litter size and the data suggest that these developmental ratios are not significantly reduced at least in vivo. we agree it is not consistent but we would like to keep the results for the record.
 2. Thank you for pointing it out, we corrected it.
 3. Thank you for your comment, we added the total number of oocytes we used in Table 2 and Table 3.
Reviewer 3 Report
Comments and Suggestions for Authors
Abstract: no comments
Introduction: in line 38, authors used a term "unexplained infertility", do they have any reference for that?
Materials and methods: no comments
Results: The way results were presented is monotonous.
Discussion: no comments
Comments on the Quality of English LanguageNo comments over English language
Author Response
 Thank you for reviewing our manuscripts. Regarding the comment on introduction, the definition of unexplained infertility means there were no currently available diagnostic methods that could detect the symptoms in both the male and female of the infertile couple. You can find the definition of unexplained infertility as well as the current status of in the reference number 5. I made some changes in the introduction (Lines 38-40).
Round 2
Reviewer 2 Report
Comments and Suggestions for Authors
The authors have satisfactorily addressed the concerns previously raised. The revisions enhance the clarity and robustness of the manuscript.